# Unraveling the energy storage mechanism in graphene-based nonaqueous electrochemical capacitors by gap-enhanced Raman spectroscopy

Xiao-Ting Yin[1], En-Ming You[2], Ru-Yu Zhou[1], Li-Hong Zhu[3], Wei-Wei Wang[1], Kai-Xuan Li[1], De-Yin Wu [1], Yu Gu [1] ✉, Jian-Feng Li [1] ✉, Bing-Wei Mao[1] & Jia-Wei Yan [1] ✉

Graphene has been extensively utilized as an electrode material for nonaqueous electrochemical capacitors. However, a comprehensive understanding of the charging mechanism and ion arrangement at the graphene/electrolyte interface remain elusive. Herein, a gap-enhanced Raman spectroscopic strategy is designed to characterize the dynamic interfacial process of graphene with an adjustable number of layers, which is based on synergistic enhancement of localized surface plasmons from shell-isolated nanoparticles and a metal substrate. By employing such a strategy combined with complementary characterization techniques, we study the potential-dependent configuration of adsorbed ions and capacitance curves for graphene based on the number of layers. As the number of layers increases, the properties of graphene transform from a metalloid nature to graphite-like behavior. The charging mechanism shifts from co-ion desorption in single-layer graphene to ion exchange domination in few-layer graphene. The increase in area specific capacitance from 64 to 145 $\mu$F cm$^{-2}$ is attributed to the influence on ion packing, thereby impacting the electrochemical performance. Furthermore, the potential-dependent coordination structure of lithium bis(fluorosulfonyl) imide in tetraglyme ([Li(G4)][FSI]) at graphene/electrolyte interface is revealed. This work adds to the understanding of graphene interfaces with distinct properties, offering insights for optimization of electrochemical capacitors.

The pursuit of energy storage and conversion systems with higher energy densities continues to be a focal point in contemporary energy research. electrochemical capacitors represent an emerging class of electrochemical energy devices that bridge the gap between conventional capacitors and batteries. They amalgamate the high-power attributes of conventional capacitors with the high-energy characteristics inherent to batteries[1]. The remarkable power-handling capabilities and extended cycling lifespan of electrochemical capacitors come at the cost of reduced energy density. Recently, material-focused research has been carried out to mitigate this trade-off by fine-tuning the performance of carbon electrodes[2,3], and by incorporating ionic liquid electrolytes engineered to withstand higher voltages[4].

[1]State Key Laboratory of Physical Chemistry of Solid Surfaces, College of Chemistry and Chemical Engineering, Xiamen University, Xiamen, China. [2]School of Ocean Information Engineering, Fujian Provincial Key Laboratory of Oceanic Information Perception and Intelligent Processing, Jimei University, Xiamen, China. [3]Department of Electronic Science, Xiamen University, Xiamen, China. ✉e-mail: ygu@xmu.edu.cm; li@xmu.edu.cn; jwyan@xmu.edu.cn

Both strategies have achieved notable improvements in energy density while preserving power density.

Graphene is a promising carbon material for use as an electrode in electrochemical energy storage devices due to its stable physical structure, large specific surface area ($\sim 2600\ m^2 \cdot g^{-1}$), and excellent electrical conductivity[5]. The first report on the use of graphene as an electrode material for electrochemical capacitors was published in 2008[6], showing the great potential of its application in electrochemical storage devices. In the realm of electrochemical capacitor applications, graphene materials present distinctive advantages. Their outstanding specific surface area enables the attainment of higher specific capacitance and energy storage density. In addition, their exceptional electrical and thermal conductivity work synergistically to diminish internal resistances within capacitors, thereby elevating charge-discharge rates and power densities. The layered structure facilitates electrolyte wetting and ion adsorption/desorption, while the stable stacking of layers effectively leverages double layer surface area to augment capacitance, thus facilitating ion diffusion[7]. Single-layer graphene serves as a typical zero-bandgap material, which exhibits a behavior mechanism similar to that of graphite as the number of layers increases[8–11]. This evolution from metalloid to graphitic-like attributes triggers consequential changes in the chemical nature of graphene, thereby exerting a pronounced influence on the ensuing electrochemical behavior at interfaces. On the other hand, the exploration of innovative electrolytes plays a pivotal role in enhancing electrochemical capacitor performance, encompassing the electrochemical window, safety, and adaptability in broad temperature ranges[12]. Solvate ionic liquids stand out by not only harnessing the advantages of ionic liquids but also leveraging the introduction of alkali metal ions to significantly boost system conductivity[13,14]. Lithium bis(fluorosulfonyl) imide (LiFSI) with a weakly associated anion can promote ion dissociation. Glyme molecules ($CH_3O(CH_2\text{-}CH_2O)_nCH_3$), serving as multidentate ligands, create stable complexes with alkali metal cations in electrolytes. These complexes contribute to high conductivity, low viscosity, and low melting points in the system, thereby further reducing impedance and enhancing rate capability[15]. Accurately revealing the graphene/solvate ionic liquid interface can provide profound insights into interfacial behavior, which benefits understanding the energy storage mechanism and guiding the design of graphene-based nonaqueous electrochemical capacitors.

The mechanism of charge storage in electrochemical capacitors has traditionally been attributed to the electrosorption of ions on the surface of a charged electrode to form an electrical double layer[16]. Nevertheless, contemporary empirical observations have unveiled a more intricate mechanism, wherein factors such as relative pore/ion sizes[17,18], along with desolvation effects[19,20], play pivotal roles. Numerous inquiries concerning the charging mechanism in practical devices persist without clear answers. In particular, how the configuration of adsorbed ions influences the capacitance, or whether the exchange of ions with opposite charges, coupled with the ejection of co-ions from charged electrodes, contributes to the formation of the electrical double layer. Therefore, exploring the charging mechanism of the electrical double layer and understanding its relationship with the electrochemical performance remains a challenge so far. Although the charging mechanism has been intensively investigated by various advanced experimental techniques[1,21] and theoretical simulations[22,23], the charge/ion separation mechanism during the dynamic process of interfacial polarization at the graphene/electrolyte interface is still not well understood, hindering the large-scale application of graphene materials in electrochemical capacitor devices.

Raman spectroscopy stands as a powerful technique for in-situ characterization of electrochemical interfaces at the molecular level. Nevertheless, the plasmon resonance at graphene surfaces lies beyond the visible light range, thereby precluding electromagnetic field enhancement. Meanwhile, its chemical enhancement factor hovers around $10^2$, thus rendering Raman signals insufficient[24,25]. Therefore, obtaining Raman signals at graphene surfaces remains a challenge. Shell-isolated nanoparticle-enhanced Raman spectroscopy (SHINERS) has emerged as one of the most effective methods for probing interfacial processes. For example, it has been extensively applied to investigate interfacial reactions in aqueous solutions[26,27]. The principle of SHINERS is to coat an ultra-thin ($\sim 2\ nm$ thick) and inert shell on the surface of metal nanoparticles with surface-enhanced Raman spectroscopy (SERS) activity, whose enhancement is mainly from the enhanced near electric field generated by surface plasmon (SP), which is the collective oscillation of free electrons in plasmonic nanomaterials such as gold, silver, and copper under the excitation of electromagnetic radiation. The SHINERS avoid direct contact of the measured species with plasmonic nanomaterials and allow for tracking an interfacial process on any substrate surface[28,29]. Recently, we developed a depth-sensitive plasmon-enhanced Raman spectroscopy (DS-PERS) method based on SHINERS to monitor and elucidate the process of sequential formation of solid-electrolyte interphase (SEI) on the Cu current collector and then on the freshly deposited Li[30]. Unfortunately, the enhancement of the Raman signals on graphene is too weak to directly obtain high-quality Raman signals at the graphene/electrolyte interface, even using SHINERS.

Herein, we design a gap-enhanced Raman spectroscopy strategy for studying the behavior and mechanism of the graphene/electrolyte interface. A sandwich configuration is employed for achieving the enhancement using the localized surface plasmon (LSP) effect coupling between shell-isolated nanoparticles (SHINs) and metal substrate. The bottom layer of the configuration is Au substrate, the top layer is Au@SiO2 nanoparticles, and the middle layer is graphene with an adjustable number of layers. With the LSP effect, the Raman signal of the measured species at the graphene surface is enhanced. Solvate ionic liquid ([Li(G4)][FSI]) is used as the electrolyte to understand the adsorption behavior in the electrical double layer of graphene/[Li(G4)][FSI] interface and the potential dependence of different coordination structures of [Li(G4)][FSI] at the interface. By modulating the number of graphene layers, a transition from metalloid to graphite-like can be achieved. Through the combination of in-situ Raman spectroscopy with electrochemical techniques, the interface of graphene/electrolyte with diverse properties was explored. Our investigations unveil significant dependencies of ion adsorption configurations and mechanisms on the number of graphene layers. These findings deliver comprehensive insights into the adsorption and exchange processes of ions on graphene electrodes, thereby offering novel avenues for comprehending and manipulating the charging dynamics of electrochemical capacitors.

## Results and discussion

### Fundamental concept of gap-enhanced Raman spectroscopy

The sandwich configuration of the Au substrate coupled with shell-isolated nanoparticles to enhance the Raman signal is shown in Fig. 1a, which is abbreviated as Au@SiO2/graphene/Au. Graphene was transferred onto a flat Au substrate, and then Au@SiO2 nanoparticles were transferred on the surface of the graphene to construct an Au@SiO2/graphene/Au electrode. SEM image of SHINs on graphene is shown in Fig. S1. In addition, by manipulating the growth time of graphene on a copper substrate using the Chemical Vapor Deposition (CVD) method, accurate control over the number of graphene layers can be achieved. With the increase of the number of graphene layers, the property of graphene could gradually transform from metalloid to graphite-like. Within this specific configuration of coupling with the metal substrate, under appropriate laser irradiation, highly SERS-active Au nanoparticles coated with an ultra-thin ($\sim 2\ nm$ thick) $SiO_2$ shell can generate an exceptionally strong electromagnetic field, which amplifies the Raman signals of adsorbed species at the graphene interface. Consequently, the potential-

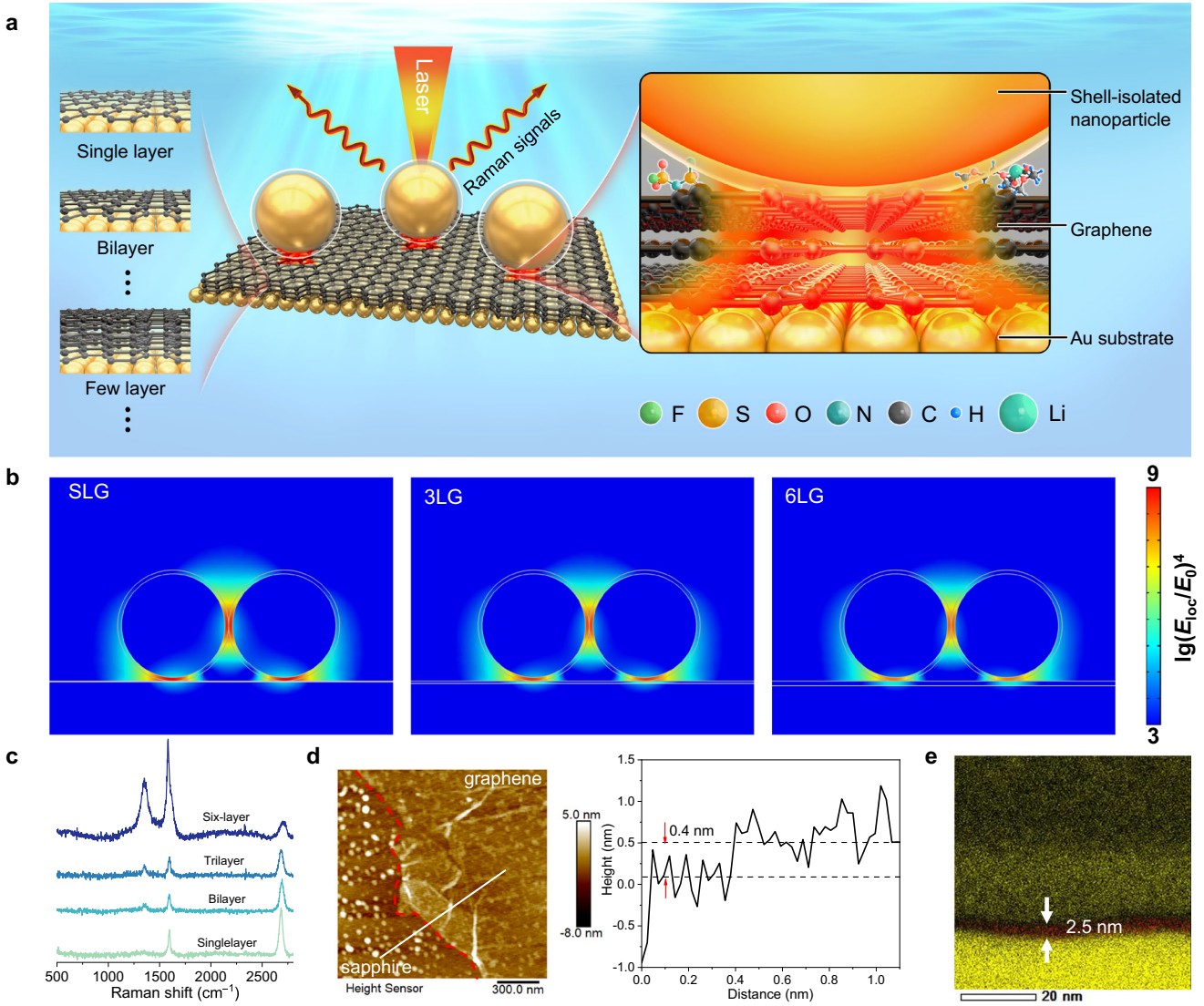

**Fig. 1 | Au@SiO₂/graphene/Au configuration. a** Sandwich configuration of Au substrate coupled with shell-isolated nanoparticles for enhancing Raman signals. F, S, O, N, C, H, and Li atoms are shown in green, yellow, red, cyan, gray, blue, and reseda, respectively. **b** Electromagnetic field distributions around Au@SiO₂/graphene/Au simulated by COMSOL finite element method. **c** Raman spectra of graphene with different layers. **d** AFM image of single-layer graphene on a sapphire substrate (with the red dashed line indicating the boundary) and sectional analysis of the white line in the image for measuring the height of the transferred graphene. Scale bar: 300 nm. **e** TEM mapping analysis of Au@SiO₂/graphene/Au. C and Au are shown in red and yellow, respectively. Scale bar: 20 nm.

dependent information of adsorbed species at the graphene interface can be obtained.

In order to verify the feasibility of the sandwich configuration of graphene with different layers, the COMSOL finite element method was used to simulate the electromagnetic field distribution of the configuration. The results are shown in Fig. 1b. It is evident that the most intense region of Raman scattering enhancement, commonly referred to as the hotspot, is localized precisely at the interface between the nanoparticles and the graphene. The enhancement factor for the single-layer graphene (SLG) configuration can reach the order of 109, while for the three-layer graphene (3LG) configuration, the enhancement factor can reach the order of 108. Even in the case of a six-layer graphene (6LG) configuration, the enhancement factor can still reach the order of $10^6$ or higher, demonstrating the synergistic interplay of electromagnetic field enhancement and chemical enhancement. It should be pointed out that six-layer graphene exhibits a behavior similar to graphite[10]. Similar enhancement was observed on the Cu substrate (Figure S2). Building upon the aforementioned, the challenge of weak coupling between graphene and shell-isolated

nanoparticles is effectively addressed by the sandwich configuration (Figs. S3–5), which can be used to investigate the graphene/electrolyte interface and track the displacement of all electrolyte species including solvent molecules, thus provide a comprehensive description of the structure of the electrical double layer.

The sandwich configuration was characterized by Raman spectroscopy, transmission electron microscopy (TEM), and atomic force microscopy (AFM). Figure 1c shows the Raman spectra of graphene under the laser irradiation of 532 nm. Compared with SLG, the increase in the number of layers results in a much broader and up-shift 2D band. The further increase in the number of layers leads to a significant decrease in the relative intensity of the 2D peak[31]. The results show that SLG, bilayer, and few-layer graphene have been successfully transferred to Au substrates. As shown in AFM images (Fig. 1d and Figure S6), graphene with different layers is observed on the smooth substrate. The measured height of single and six-layer graphene is about 0.4 and 2.5 nm, respectively. The sandwich configuration was further characterized by TEM (Fig. 1e), with Au@SiO₂ nanoparticles positioned above the graphene and Au substrates situated below the

graphene. The size of the gap between Au@SiO$_2$ nanoparticles and Au substrate is about 2.5 nm (~ 0.4 nm for a single-layer)[32], indicating that the six-layer graphene is successfully inserted into the nanogap.

## Electrochemical techniques and in situ AFM

The chemical structure of lithium bis(fluorosulfonyl) imide in tetraglyme ([Li(G4)][FSI]) is shown in Fig. 2a. Cyclic voltammetry measurements of graphene in [Li(G4)][FSI] were conducted with a scan rate of 10 mV s$^{-1}$ between 2.1 and 3.9 V (vs. Li$^+$/Li) in a three-electrode cell, showing capacitive charge-discharge behavior (Fig. 2b). By comparing with the CV of Highly Oriented Pyrolytic Graphite (HOPG) in [Li(G4)][FSI], the metalloid to graphite-like properties of the graphene with tunable number of layers were confirmed (Figure S7). Metalloid SLG and 3LG can strongly adsorb anions and cations, resulting in hysteresis behavior in the CV curves, while the adsorption of anions and cations on graphite-like 6LG is weak, and thus the hysteresis behavior disappears, indicating that the number of graphene layers can affect the adsorption kinetics in electrical double layer. Ion adsorption on metal electrodes is much stronger than on carbon electrodes, leading to a quite different electrical double layer structure[33].

As a two-dimensional nanomaterial, graphene presents quantum capacitance (QC)[34]. The differential capacitance curve of graphene electrodes with different layer numbers is shown in Fig. 2c, and the Nyquist and complex capacitance diagrams are shown in Figures S8–10. With the increase in the number of graphene layers, the differential capacitance curve changes from U-shape to V-shape. For single-layer graphene, the quantum capacitance plays a dominant role, thus the total capacitance behavior of the interface is similar to the quantum capacitance behavior. The potential of zero charges (PZC) corresponds to the potential with the minimum differential capacitance[35]. The total differential capacitance curve exhibits symmetry around the PZC, both in the positive and negative directions. For few-layer graphene (6LG), electrical double layer capacitance takes a dominant role, with a higher quantity of anions adsorbed at the interface in comparison to cations. As a result, the differential capacitance curve exhibits a characteristic left-low, right-high profile. The differential capacitance curve for HOPG is depicted in Figure S11, showing a distinctive V-shape. Our results are supported by the experiments for the graphene/ionic-liquid interface, which show that total capacitance is limited by quantum capacitance when the number of graphene layers (N) is less than 4 and by EDL capacitance when N > 4[36,37]. At the same time, it can be found that the differential capacitance increases with the number of graphene layers. At the metalloid single-layer graphene interface, the robust charge-charge interactions prompt cations and anions to establish Coulombic ordering and densely ion arrangements. Disrupting this ordered structure requires substantial polarization (E < 2.5 V or E > 3.5 V), consequently leading to a reduction in capacitance. Conversely, at graphite-like few-layer graphene interface, the charge-charge interactions are less pronounced, leading to reduced Coulombic ordering and more loosely packed ion configurations. This facilitates the separation of cations and anions under low polarization, resulting in a higher capacitance. The electrochemical capacitors utilizing few-layer graphene with an ABA stacking structure can achieve higher double layer capacitance compared to single-layer graphene. This occurrence is attributed to the increased intensity of image forces in ABA stacking, which disrupts ionic ordering and facilitates the formation of effective free ions[38].

We further investigated the nanostructures of anions and cations at the interface by atomic force microscopy (AFM), which is a powerful method to detect the ion arrangement and layered structure of electrical double layers in ionic liquids[39–41]. Figure 2d, e and Figure S12, 13 show potential-dependent force-distance curves. It can be seen that there exists a multilayer interfacial structure at the nanometer scale from the electrode surface, which gradually attenuates at the electrolyte side. In Fig. 2d, the EDL of this system consists of two layers of

nanostructures, the measured thicknesses of the two layers are 0.52 and 0.32 nm, respectively. The corresponding layer thicknesses match the size of Li(G4)$^+$ and FSI$^-$ ions[42]. Therefore, it is reasonable to infer that the negatively charged electrode leads to the enrichment of Li(G4)$^+$ cations in the innermost layer, and the second layer is composed of FSI$^-$ anions. When the electrode is positively charged, the thickness of the first layer decreases from 0.52 to 0.32 nm, and the thickness of the second layer increases from 0.32 to 0.52 nm, which demonstrates that the innermost layer of the EDL is enriched with FSI$^-$ anions, and the second layer is enriched with Li(G4)$^+$ cations. The above show that the thickness of the electrical double layer is within 1 nm, the ultra-thin (1 nm thick) electrical double layer benefits obtaining higher specific capacitance.

## Distinguishing charge storage mechanisms

Although AFM reveals the potential-dependent rearrangement of cations and anions, and the number and thickness of the layered structure at the interface, it lacks chemical sensitivity. In order to further understand the energy storage mechanism of the electrical double layer at the molecular level, Raman spectra of the electrode/[Li(G4)][FSI] interface were obtained using the Au@SiO$_2$/graphene/Au sandwich configuration. For single-layer graphene, in-situ Raman measurements were performed in a wide potential region from 2.1 to 3.7 V (Fig. 3a). We opted for a 785 nm laser wavelength to investigate the adsorption configuration of ions and avoid fluorescence interference. In the very low energy regime, the intensity of the G and 2D peaks is suppressed due to the conservation of angular momentum associated with continuous rotational symmetry in the low-energy regime[43]. Consequently, the G and 2D peaks were not observed. The spectral peaks at 260 and 293 cm$^{-1}$, which are assigned respectively to the rocking of S-F and SO$_2$[44], exhibit potential dependence and a notable inflection point at 3.0 V (vs. Li$^+$/Li), indicating that the PZC is around 3.0 V, being close to the aforementioned potential with the minimum differential capacitance. At potentials positive of the PZC (Fig. 3a, blue), the intensity of the rocking of SO$_2$ at the interface is stronger than that of SF, which is similar to the bulk spectrum. It is referred that FSI$^-$ anions are lying flat on the graphene. While FSI$^-$ ions adjacent to the graphene maintain a parallel arrangement with respect to the electrodes surface in N-down configuration (Fig. 3b), Li(G4)$^+$ ions also maintain a parallel arrangement with respect to the electrodes to maximize the Coulombic interaction with the paralleled aligned FSI$^-$ ions. The band at 1400–1500 cm$^{-1}$ represents the CH$_2$ bending/scissoring mode of G4[45], the peak exhibits a blueshift corresponding to the positive shift of the potential, which suggests that Li(G4)$^+$ cations are pushed away from the electrode surface, as shown in Figure S14. A portion of Li(G4)$^+$ rearranges in the vertical direction and desorbs from the electrode.

However, at potentials negative of the PZC (Fig. 3a, purple), the peak intensity of the rocking of SO$_2$ at the interface is weaker than that of the rocking of SF, indicating that FSI$^-$ anions are pushed away from the electrode surface. The peak at 868 cm$^{-1}$ represents the breathing mode of crown ether-like Li(G4)$^+$[46], which appears at 3.3 V and then shifts to 874 cm$^{-1}$ as the potential decreases to 2.7 V, as shown in Figure S15. Meanwhile, for the two peaks at 930 and 1138 cm$^{-1}$ being the coupling mode of CH$_2$ bending, CO stretching, and CC stretching vibration[47], the peak of CH$_2$ bending/scissoring mode of G4 undergoes a redshift in correlation with the negative shift of the potential. Moreover, Li(G4)$^+$ is absorbed on the surface lying flat, which further shows that the FSI$^-$ ions are rearranged and desorbed from the electrode, and there is an increase in the intensity of Li(G4)$^+$ ions in the first ionic layer. That is the rearrangement of the ionic layer due to the polarization of the electrode is accompanied by the transition of the ions near the electrode. Thus, at the metalloid interface, the charging mechanism is that within the electrical double layer adjacent to the electrode, the desorption of the co-ion with charge being the same as

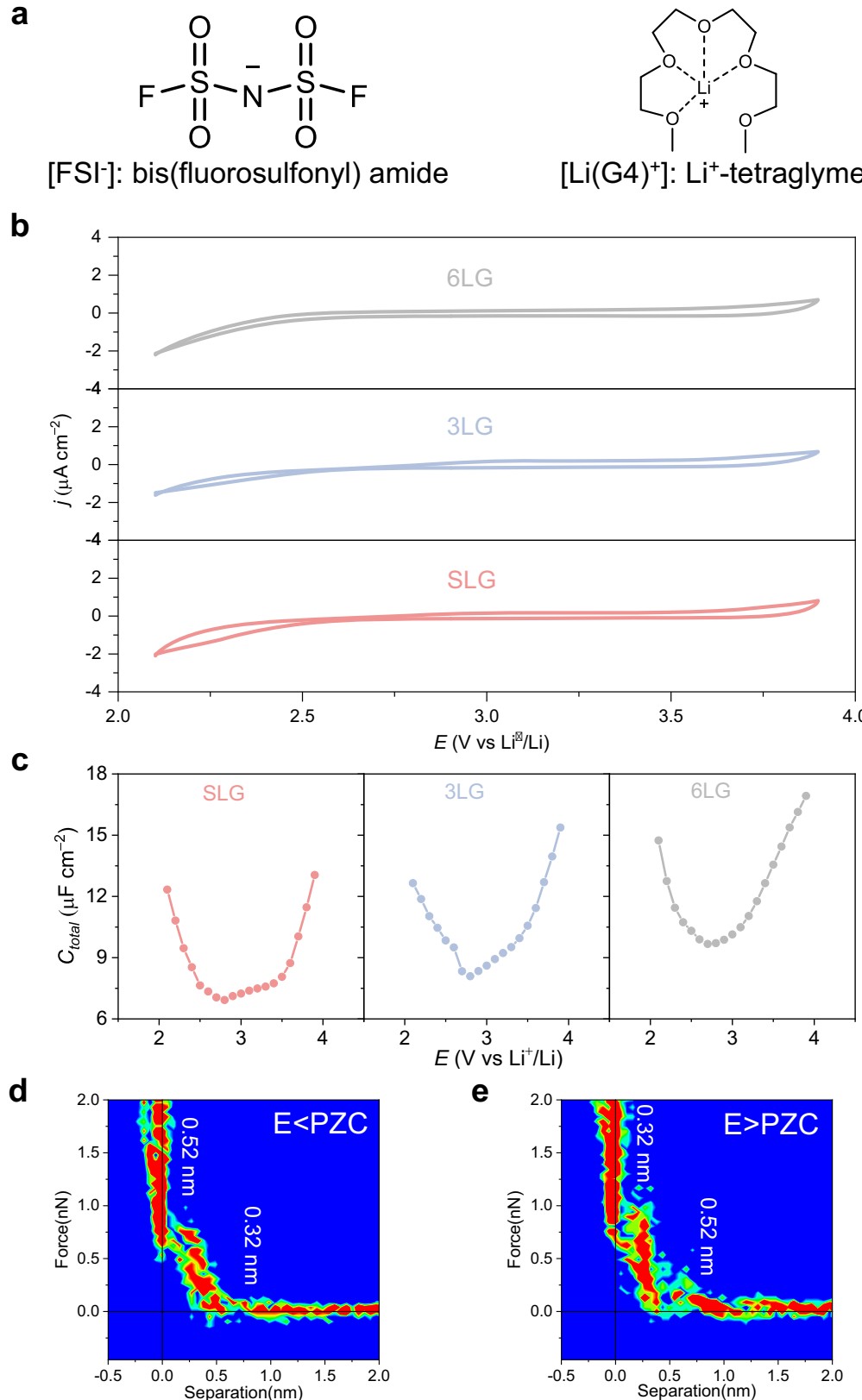

**Fig. 2 | Electrochemical techniques and in situ AFM characterization of graphene with different layers. a** Chemical structures of anion and complex ion used in this work. **b** Cyclic voltammograms of SLG (pink), 3LG (blue), and 6LG (gray) electrodes plotted with respect to electrode potentials versus Li⁺/Li. Scan rate: 10 mV s⁻¹. **c** Differential capacitance curves of SLG (pink), 3LG (blue), and 6LG (gray) electrodes plotted with respect to electrode potentials versus Li⁺/Li. **d**, **e**, Two-dimensional graphs of AFM force-distance profiles from 20 force curves under E < PZC and E > PZC. E, electrode potential. PZC, the potential of zero charges. SLG, single-layer graphene; 3LG, three-layer graphene; 6LG, six-layer graphene.

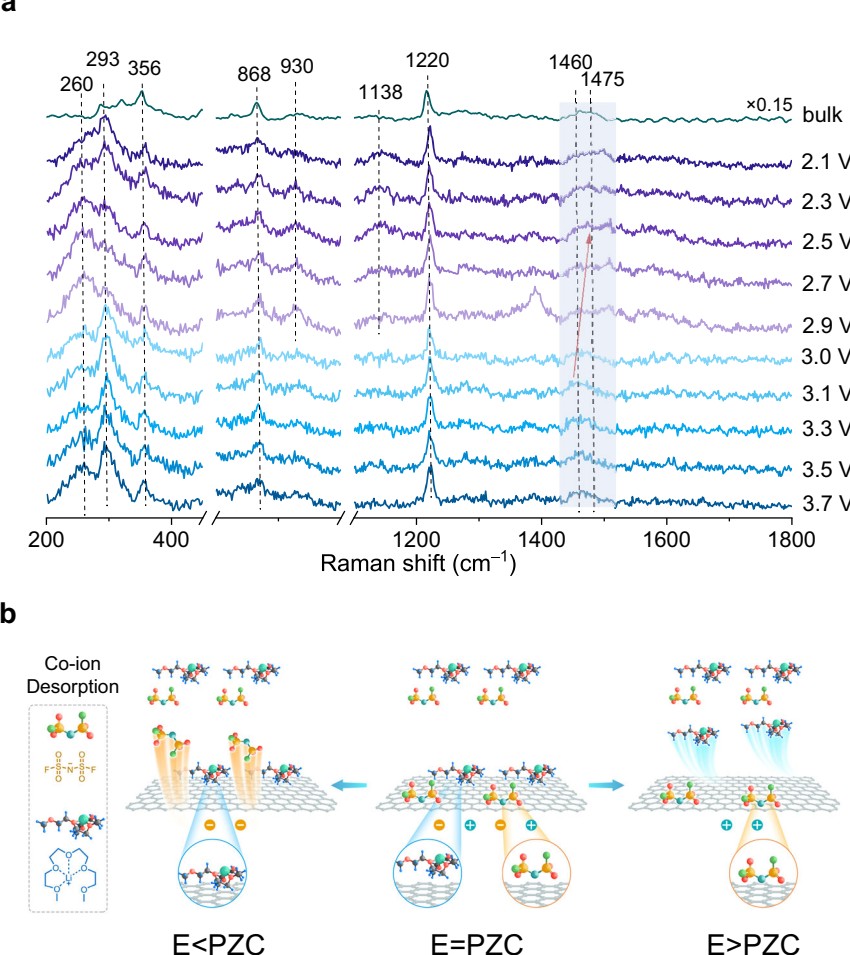

**Fig. 3 | In situ Raman results of [Li(G4)][FSI] on single-layer graphene surface.** **a** In-situ Raman spectra of [Li(G4)][FSI] at Au@SiO$_2$/single-layer graphene/Au. The purple Raman spectra correspond to the potential being negative relative to PZC, while the blue Raman spectra correspond to the potential being positive relative to PZC. **b** Proposed adsorption configuration of [Li(G4)][FSI] at single-layer graphene in different potential region (including E < PZC, E = PZC and E > PZC). F, S, O, N, C, H, and Li atoms are shown in green, yellow, red, cyan, gray, blue, and reseda, respectively.

the electrode is more pronounced than the adsorption of counter-ion with charge opposite to the electrode. Regarding the strong vibration peak appearing at 2.9 V at ~1400 cm$^{-1}$, it is ascribed to the asymmetry stretching of SO$_2$. When the potential shifts from negative to positive relative to PZC, the interface undergoes a transition from FSI$^-$ co-ion desorption to Li(G4)$^+$ co-ion desorption. The alteration in the configuration of FSI$^-$ leads to the observed SO$_2$ asymmetry stretching. It should be noted that when using the Au@SiO$_2$/graphite configuration, the adsorption behavior of the above electrolyte components at the electrochemical interface cannot be observed, as graphite could not provide electromagnetic field enhancement and chemical enhancement factor is only 10$^2$, as shown in Figures S16, 17.

By adjusting the thickness of graphene, the interface between graphene and [Li(G4)][FSI] is further explored to understand the behavior of graphene as a host material in electrochemical capacitors. As shown in Figure S18, when the number of graphene layers increases to three, the potential-dependent behavior of FSI$^-$ and Li(G4)$^+$ can still be observed at the hot spot, which is similar to the interfacial behavior of single-layer graphene. However, when the number of graphene layers increases to six, the spectra present different characteristics as shown in Fig. 4a. At potential around the PZC, the peak intensity of SO$_2$ is nearly equivalent to that of SF, indicating that the N atoms in the anions are not close to the electrode surface, which is an N-up configuration, and that both anions and cations adsorb on graphene

surface (Fig. 4b). With positive shift of the potential, the SF peak intensity increases while the SO$_2$ peak intensity decreases, which is ascribed to the orientation change of the absorbed FSI$^-$ anions from flat adsorption to oblique adsorption and the increase of the number of anions accommodated at the interface. The Raman signal of the CH$_2$ bending/scissoring mode of G4 (1400–1500 cm$^{-1}$) becomes weak until it disappears, indicating that Li(G4)$^+$ cations move away from the interface and the number of cations decreases. At potentials negative of PZC, the peak intensity of SF decreases while the peak intensity of SO$_2$ increases, i.e., the configuration of the anion undergoes inversion. The Raman signal reduces as the potential shifts negatively, and the FSI$^-$ ions transfer until they move away from the interface. The Raman signal of the CH$_2$ bending/scissoring mode of G4 (1400–1500 cm$^{-1}$) increases with the negative shift of potential, showing that the number of cations increases and Li(G4)$^+$ cations adsorb on the surface lying flat.

The six-layer graphene exhibits distinct D-band and G-band observed at the interface and both of them present potential-dependent behavior. The D band is located at 1350 cm$^{-1}$ which is a defect-activated double-resonance Raman process[31], and the G band is located at 1580 cm$^{-1}$ which is caused by the in-plane vibration of sp$^2$ carbon atoms[48]. When E < PZC, the intensity of the D-band gradually increases because the cations adsorb on the electrode surface (Fig. 4c). However, the change of adsorbate on graphene causes the shift of the G band to present a potential dependence, and anions and

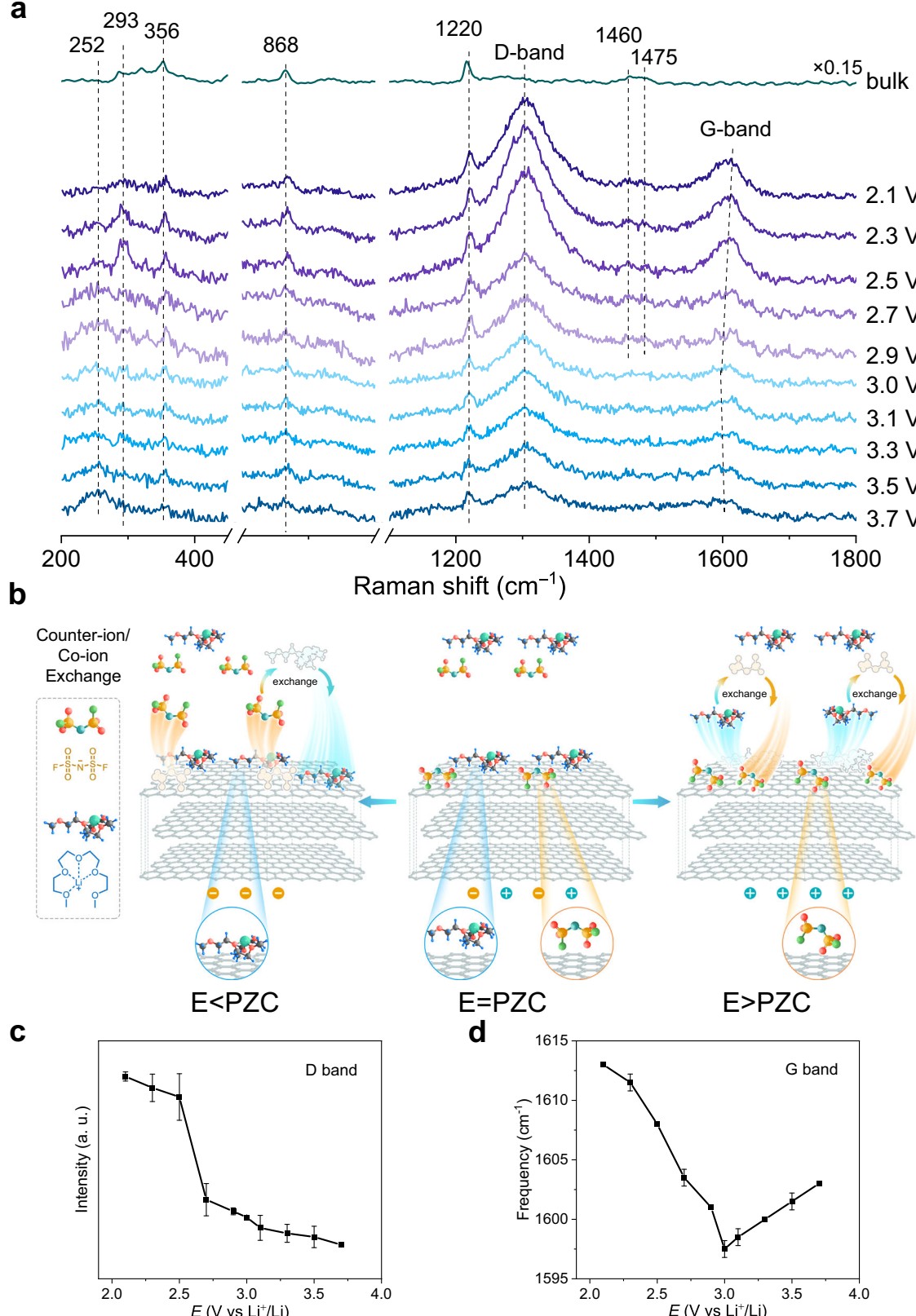

**Fig. 4 | In situ Raman results of [Li(G4)][FSI] on six-layer graphene surface. a** In-situ SHINERS spectra of [Li(G4)][FSI] at Au@SiO₂/six-layer graphene/Au. The purple Raman spectra correspond to the potential being negative relative to PZC, while the blue Raman spectra correspond to the potential being positive relative to PZC. **b** Proposed adsorption configuration of [Li(G4)][FSI] at six-layer graphene in different potential region (including E < PZC, E = PZC and E > PZC). F, S, O, N, C, H, and Li atoms are shown in green, yellow, red, cyan, gray, blue, and reseda, respectively. **c** Potential-dependent normalized intensity of the D band of graphene. **d** Potential-dependent frequency of the G band of graphene. Error bars represent s.d. for each data point ($n$ = 3 independent experiments), and points are average values.

cations adsorb on the surface as electron acceptor dopants[49,50], as shown in Fig. 4d. At a negatively charged interface, the G-band undergoes a gradual blueshift from ca. 1598 to 1613 cm⁻¹ as the potential decreases to 2.1 V, with Li(G4)⁺ cations migrating from the bulk to the interface. As the surface becomes neutral or positive, the G-band begins to significantly broaden, and the redshift is observed with a positive shift of potential since the Li(G4)⁺ cations orient away from the surface and towards the bulk while the FSI⁻ anions arrange at the interface. This shows that charge storage at the graphite-like interface is actually driven by ion exchange, whereby counter-ions are adsorbed to the interface while co-ions are simultaneously ejected, which is significantly different from the behavior of single-layer graphene metalloid interfaces.

The above experiments show that the sandwich configuration of Au substrate coupled with shell-isolated nanoparticles to enhance Raman signals can indeed obtain information about graphene with different layers at the molecular level from the graphene/electrolyte interface, thus providing an effective method for in-depth research on the electrochemical energy storage mechanism of graphene-based electrochemical capacitors. Figure S19 shows the charge/discharge behavior of single-layer/single-layer, single-layer/six-layer, and six-layer/six-layer at 10 A g⁻¹ in the potential range from − 0.8 to 0.8 V, respectively. The capacitance is estimated using the following formula ($C_s = \frac{i \cdot \Delta t}{\Delta V \cdot S}$). The area-specific capacitance of single-layer/single-layer is 64 μF cm⁻², which is mainly dominated by the charging mechanism of co-ion desorption. With the increase of the number of layers, the area-specific capacitance of six-layer/six-layer is 145 μF cm⁻², which is turned into the charging mechanism dominated by ion exchange. The relationship between the charging mechanism and the electrochemical performance has been correlated. Charging via co-ion desorption is anticipated to alleviate the enthalpic penalty associated with interactions among charges, thereby simultaneously enhancing entropy. Consequently, co-ion desorption should optimize capacitance[51]. The ion exchange mechanism, in principle, mitigates the enthalpic penalty linked to denser ion packing by maintaining a relatively constant total density throughout charging. Simultaneously, the entropic penalty associated with charging is also reduced[52]. We conducted a further comparison of the energy density of electrochemical capacitors operating under two predominant mechanisms, as shown in Table S1. Ion exchange plays a significant role in maximizing ion packing, consequently influencing the charging rate and capacitance. The above results provide a direct insight into the charge storage process in graphene electrodes with different layer numbers at the molecular level, showing that the charging mechanism depends on the layer numbers of graphene.

## Revealing coordination structures at the interface

In order to further increase the energy density of electrochemical capacitors, as a type of new capacitor-hybrid electrochemical capacitors, lithium-ion capacitor has been developed in recent years[53,54], which is an electrochemical energy storage device with performance between lithium-ion batteries and electrochemical capacitors. An intercalated/deintercalated lithium-ion electrode material and an electrical double layer capacitor electrode material with a large specific surface area are used for the two poles, respectively. The energy storage mechanism includes both the intercalation/deintercalation of lithium ions in the electrode material and the absorption/desorption of electrolyte ions on the surface of the electrode material. Therefore, lithium-ion capacitors combine the advantages of lithium-ion batteries and electrochemical capacitors, which not only have higher power density and longer cycle life than lithium-ion batteries but also have higher energy density than electrochemical capacitors. Solvate ionic liquid is a promising class of electrolyte for lithium-ion capacitors. LiFSI exhibits weak ion-ion interactions, while G4 provides strong ion-

solvent interactions, which collectively influence the desolvation process, thus affecting the Li⁺ deintercalation behavior on the electrode surface. When LiFSI is dissolved in a solvent, Li⁺ ions interact with the solvent molecules and FSI⁻ anions, resulting in the following three types of coordination structures: Solvent-separated ion pairs (SSIP), in which only solvent molecules are present in the solvation shell around cation; Contact ion pairs (CIP), solvent molecules and one anion are present in the solvation shell around the cation to form a neutral complex; Aggregates (AGG), solvent molecules and multiple anions are present in the solvation shell around the cation[55]. The anions in the form of CIP and AGG are the main precursors of anion-derived SEI film[56]. Therefore, coordination structures in electrolytes play a vital role in dictating the components and structure of anion-derived SEI[57,58], and thus affect the performance of lithium-ion capacitors. The sandwich configuration provides the opportunity to reveal the coordination structure of [Li(G4)][FSI] at the graphene/electrolyte interface by Raman spectroscopy.

Potential-dependent Raman spectra of coordination structures at single-layer graphene/[Li(G4)][FSI] interface are shown in Fig. 5. The peaks at 700–770 cm⁻¹ are assigned to the S-N stretching vibrations of FSI⁻ anions, which are highly sensitive to the coordination environment[59,60]. For the free and uncoordinated FSI⁻ anion, SSIP (Fig. 5a, teal) has a peak at about 720 cm⁻¹. The peaks of CIP (Fig. 5a, blue) and AGG (Fig. 5a, orange) are about 730 cm⁻¹ and 745 cm⁻¹, respectively, when the lithium salt and solvent are mixed in an equal molar ratio[55]. We analyzed the peaks measured at the interface to obtain information on coordination structures. As shown in Fig. 5b, c, at potentials negative of PZC, the content of SSIP decreases and the content of AGG increases with the negative shift of potential, while at potentials positive of PZC, SSIP content increases and AGG content decreases with the positive shift of potential, which is related to the change of anion and cation adsorption at the interface. The content and frequency of ion pairs present a turning point at 3.0 V. The content of SSIP has a local minimum point at 3.0 V, while the content of CIP + AGG has a local maximum point at 3.0 V. In the double layer region, the content of CIP + AGG is higher than SSIP, indicating that the coordination structure at the interface is potential-dependent, as shown in Fig. 5d. The peak of AGG at 743 cm⁻¹ appears at 2.9 V and then shifts to 747 cm⁻¹ as the potential decreases to 2.1 V, while the peak of AGG at 741 cm⁻¹ appears at 3.1 V and then shifts to 749 cm⁻¹ as the potential increases to 3.7 V, as shown in Fig. 5a. For few-layer graphene, the dependence of the coordination structure content on the potential can also be observed, as shown in Figure S20. However, due to the weak interaction between the electrode and the electrolyte, the ion packing is relatively loose, and the multilayer structure promotes electrolyte wetting and enhances ion adsorption/desorption, so the peak shift of the coordination structure with the potential is not obvious. These changes between single-layer graphene and few-layer graphene can be distinctly observed through Raman obtained from the Au@SiO₂/graphene/Au sandwich configuration, thus revealing the coordination structure of the solvate ionic liquid at the electrode/electrolyte interface, which will greatly improve our understanding of the complicated interfacial behavior of graphene and solvate ionic liquid-based hybrid capacitors.

In summary, we have developed a gap-enhanced Raman spectroscopy strategy of utilizing the LSP effect from the coupling of SHINs with a metal substrate for graphene electrodes with adjustable layers. This method has been employed to investigate the interfacial behavior and mechanism on graphene electrodes and can be further extended to other non-metallic substrates or indirect SERS platforms. The combination of in-situ Raman spectroscopy with electrochemical techniques facilitates a deeper understanding of the charged storage mechanism of graphene with varying layers and properties in electrochemical capacitors. We monitor a change

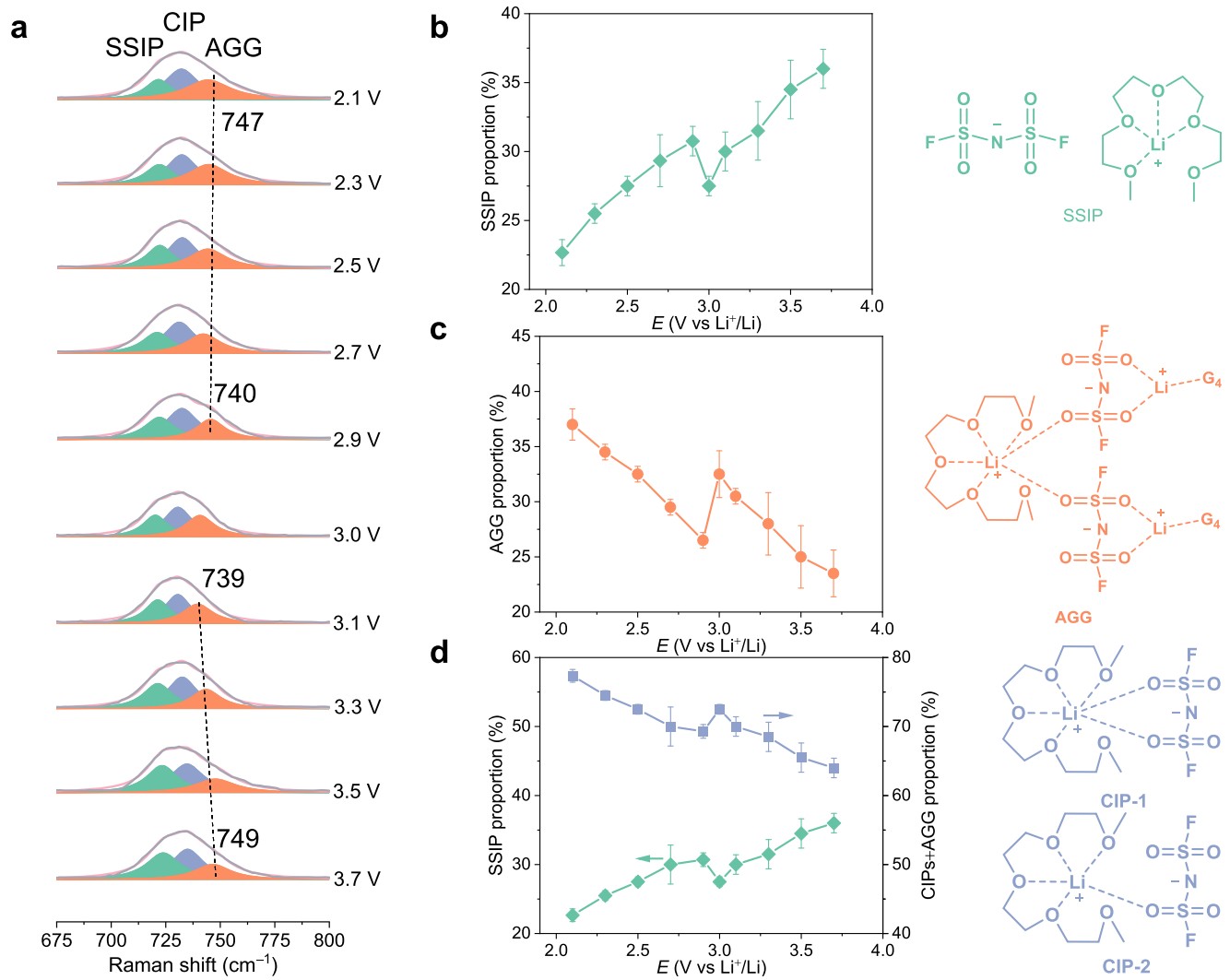

**Fig. 5 | The coordination structures at single-layer graphene/[Li(G4)][FSI] interface. a** Potential-dependent Raman spectra of coordination structures at single-layer graphene/[Li(G4)][FSI] interface. **b–d** Potential-dependent contents of SSIP (teal), CIPs (blue), and AGG (orange) at single-layer graphene/[Li(G4)][FSI] interface. SSIP, solvent-separated ion pairs; CIP, contact ion pairs; AGG, aggregates. Error bars represent s.d. for each data point (*n* = 3 independent experiments), and points are average values.

of the total capacitance curve from U-shape to V-shape, which is attributed to a decrease in quantum capacitance contribution and an increase in electrical double layer capacitance contribution, along with the variation of anions/cations configuration. The charged storage mechanisms are related to the number of graphene layers. For single-layer graphene, charging proceeds by the desorption of co-ion, whereas for few-layer graphene, co-ion/counter-ion exchange dominates. The relationship between the charging mechanism of the electrical double layer and the electrochemical performance has been correlated. The area specific capacitance of single-layer/single-layer graphene is 64 μF cm$^{-2}$, while the area specific capacitance of six-layer/six-layer graphene is 145 μF cm$^{-2}$. Furthermore, the coordination structure of [Li(G4)][FSI] at the graphene/electrolyte interface has been revealed, which indicates that the coordination structure at the interface is potential-dependent. The above finding offers new insights into the electrochemical interfaces between graphene and electrolyte. The gap-enhanced Raman spectroscopy strategy could expand the range of in-situ studies of various two-dimensional materials, including electrical double layer, electrolyte decomposition, SEI formation, and electrocatalysis.

## Methods

### Materials
Tetraglyme (G4, Santa Cruz, ≥ 99.8 %) and Lithium Bis(fluorosulfonyl) imide (LiFSI, Nippon Shokubai, ≥ 99.9 %) were mixed with a mole ratio of LiFSI: G4 = 1:1 in a glovebox filled with argon gas (VG1500, H$_2$O < 0.1 ppm, O$_2$ < 0.1 ppm), and the mixture was stirred for 24 h to obtain a homogeneous liquid at room temperature. The liquid was abbreviated as [Li(G4)][FSI], which was stored and used in a glovebox to prevent H$_2$O contamination. Chloroauric acid (99.99 %), sodium citrate (99.0 %), and (3-aminopropyl) trimethoxysilane (APTMS, 97 %) were purchased from Alfa Aesar; sodium silicate solution (27 % SiO$_2$) was purchased from Sigma-Aldrich. All chemicals were used as received without further purification.

### Synthesis of Au@SiO$_2$ shell-isolated nanoparticles
The synthesis of gold nanoparticles was carried out using the classic sodium citrate reduction method[61]. A 0.01% weight fraction HAuCl$_4$ aqueous solution (200 mL) was heated to boiling, followed by the rapid addition of a 1 % weight fraction sodium citrate aqueous solution (1.4 mL). Maintaining a gentle boil for 40 min yielded 55 nm gold nanoparticles. Taking 30 mL of the 55 nm gold nanoparticle solution,

0.4 mL of a 1 mM solution of 3-aminopropyltrimethoxysilane was added, followed by stirring for 15 min. Subsequently, 3.2 mL of a 0.54 % sodium silicate aqueous solution was added and stirring was continued for 3 min. The reaction vessel was then heated in a water bath at 95 °C, and after 20 min, core-shell particles with a shell thickness of ~ 2 nm were obtained, as depicted in Figure S1.

### Preparation of Au@SiO2/graphene/Au structure

25 µm-thick Cu foil was annealed in an $H_2$ atmosphere at 1000 °C, followed by the growth of graphene in an $H_2/CH_4$ atmosphere while maintaining elevated temperatures at 1150 °C. After completion of the growth, the system was cooled to room temperature[62]. Similarly, controlling the growth time of graphene enabled the production of multilayer graphene. A uniform layer of PMMA film was spun onto the flat graphene/Cu foil surface, cured through heating (at about 100 °C), and then subjected to ammonium persulfate etchant, causing Cu to be etched away. This yielded PMMA/graphene floating on the surface, which was meticulously rinsed with deionized water and subsequently transferred onto a Si substrate coated with 100 nm thick Au via thermal evaporation. Following drying, PMMA was removed using acetone, resulting in the graphene/Au substrate at room temperature[63]. Dispersion of SHINs in ultra-pure water led to their deposition onto the graphene/Au substrate, followed by vacuum drying. The subsequent assembly of the electrochemical cell was conducted within an argon glovebox with 0.1 ppm $H_2O$ and $O_2$ at room temperature.

### COMSOL finite element method simulation

The electromagnetic field distribution in Au@SiO2/graphene/Au configuration was calculated by COMSOL finite element analysis software. In the simulation, the diameter of Au nanoparticles was set to 55 nm, the incident laser wavelength was 785 nm, and the fine structure was $1 \times 1 \times 1$ nm$^3$.

### In situ Raman spectroscopy

Raman microscopy (Nanophoton Corporation Vis-NIR-XU) with a 785 nm laser wavelength was used to record Raman signals, and a 785 nm laser was used to avoid the fluorescent background of the electrolyte, as shown in Figure S21. A 10 mm × 10 mm graphene/Au substrate decorated with SHINs was used as the working electrode, with Li foils (purchased from China Energy Lithium Co., Ltd., ≥ 99.9 %) serving as both the reference and counter electrodes. The setup employed 500 µL of electrolyte and was assembled in an argon-filled glove box ($H_2O$ and $O_2 < 0.1$ ppm) at room temperature. All Raman measurements were performed by using a 50 × microscope objective with a numerical aperture of 0.45. The laser power was controlled at about 0.7 mW and the accumulation time was set to 60 s. The peak-versus-potential plot can be fitted with a linear equation, with the fitted line representing the slope of the Stark shift in cm$^{-1}$/V.

### AFM force spectrum measurement

All AFM (Bruker Icon) characterizations were performed in an argon-filled glovebox (MIKROUNA, $H_2O < 0.1$ ppm, $O_2 < 0.1$ ppm) equipped with an electrochemical workstation (CHI660E, Chenhua Instruments). The CSG30 probe (TipsNano) with an elastic coefficient of 0.6 N m$^{-1}$ was used to measure force curves. A freshly cleaved HOPG (20 mm 20 mm, ZYB) was used as the working electrode, and two Li wires (purchased from China Energy Lithium Co., Ltd., ≥ 99.9 %) were used as the counter electrode and the reference electrode, respectively. The setup utilized 600 µL of electrolyte. At least 20 force curves were recorded at each potential at room temperature.

### Electrochemical measurements

Cyclic voltammetry was conducted using a three-electrode configuration on an electrochemical workstation (CHI760E, Chenhua Instruments) at room temperature. A 10 mm × 10 mm graphene/ Au substrate served as the working electrode, while two lithium strips (purchased from China Energy Lithium Co., Ltd., ≥ 99.9 %) were employed as the counter electrode and the reference electrode, respectively. The volume of electrolyte used was 600 µL. All potentials stated in the study were with respect to Li$^+$/Li reference. Electrochemical impedance spectroscopy (EIS) tests were carried out on an electrochemical workstation (PGSTAT128N, Metrohm). The frequency ranged from 100 kHz to 1 Hz with an amplitude of 10 mV, and the equivalent circuit employed was R-(R)(P)-(R)(P)[35,64].

## Data availability

The data that support the findings of this study have been included in the main text and Supplementary Information. All other relevant data supporting the findings of this study are available from the corresponding authors upon request.

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

## Acknowledgements

This research was financially supported by the National Natural Science Foundation of China (22021001 (J.F.L.), 22072123 (J.W.Y.), T2293692 (J.F.L.), 22202162 (E.M.Y.), 21925404 (J.F.L.), 22372140 (J.W.Y.)) and China Postdoctoral Science Foundation (2022M722648 (E.M.Y)).

## Author contributions

J.W.Y., Y.G., and X.T.Y. conceived the idea. X.T.Y., J.W.Y., and Y.G. designed experiments and analyzed results. E.M.Y. conducted the COMSOL simulations. R.Y.Z. and L.H.Z. performed experiments. J.W.Y., J.F.L., Y.G., and X.T.Y. wrote the manuscript. W.W.W., K.X.L., D.Y.W., and B.W.M. helped with the discussion. All authors discussed and analyzed the data.

## Competing interests

The authors declare no competing interests.
