## [Peer Review File · Nature Communications]

REVIEWER COMMENTS

Reviewer #1 (Remarks to the Author):

In this manuscript the authors designed a gap enhanced Raman spectroscopic strategy to characterize the dynamic interfacial change of graphene to understand the charging mechanism and ion arrangement at the graphene/electrolyte interface. By employing such a strategy combined with complementary characterization techniques, the authors delve into potential-dependent configuration of adsorbed ions and capacitance curves for graphene with various numbers of layers. It was found that the charging mechanism shifts from co-ions desorption on single-layer graphene to co-ions/counter-ions exchange domination on few-layer graphene. Generally, the manuscript is well organized but there are still some concerns which need to be solved, including the logic thread, experimental design and data processing. Thus, I recommend the acceptance after the following comments are considered.

1. In the abstract or the introduction, the authors did not mention how the transformation of the double layer charging mechanism affects the electrochemical performance. For example, values for the specific capacitance per area under different situations and the possible reasons could be summarized.
2. The full names of the abbreviations appeared for the first time in the abstract such as [Li(G4)] and [FSI] should be given. The reason to choose this electrolyte for the research may be explained.
3. In Figure.1d, how AFM image confirms that the transferred sample is single-layer graphene is not clear enough.
4. In Figure.2b, the numerical value of the current density displayed on the vertical axis is not clear enough. It is suggested to divide it into three charts.
5. In Figure.2d and 2e, the authors stated that the thickness in the Columbic ordering layers is 0.52nm and 0.32nm, respectively, but it is difficult to distinguish two peaks from the figures. The positions may be indicated more clearly. The statistical data on multiple graphene samples for the measurement may help to determine the error of values measured.
6. In Figure.3a, the reason for the absence of G and 2D peaks of single-layer graphene shall be explained. Readers may want to know the attribution of the strong vibration peak appearing at 2.9V at $\sim 1400\text{ cm}^{-1}$.
7. In Figure.3b, different potential regions are not reflected in the figure. It would be better if the authors could provide the values of the potential regions.
8. In Figure.3a and Figure.4b, the corresponding meanings of the blue and purple parts in the Raman spectra should be explained in the caption.

9. Is there a connection between the potential-dependent coordination structure of [Li(G4)][FSI] at graphene/electrolyte interface mentioned in the last part and the charge storage mechanism mentioned earlier? Please explain.

Reviewer #2 (Remarks to the Author):

The authors show that placing graphene and few-layer graphene over a gold substrate and then coupling with SHINERS on the carbon surface— the graphene/ electrolyte interface can be investigated in systems relevant to electrochemical capacitors under potential control.

Data shown in Figure 2 is impressive and the authors demonstrate the changing capacitance shape from U to V shape going from SLG, 3LG and 6LG. Force curves show the flipping of the ions either side of pzc. The SHINERS study does show systematic changes observed in a few “weak” peaks. Overall an interesting study – but before acceptance the authors need to clarify some of their interpretation and presentation of the data as some aspects seem unclear.

1. Is the Au substrate itself SERS active? – authors need to demonstrate lack of or presence of enhancement and provide some characterisation of the Au films.

2. Authors discuss that even with SHINERS no appreciable enhancement can be made on graphene/carbon substrates and only when Au underlayer is present this is possible. Is graphene unique in this sense requiring the Au to couple with? In the authors seminal 2010 Nature paper – they show an example of detection of pesticide on an orange, what is it about an orange substrate whereby SHINERS works, whereby graphene not so?

3. The authors state: “The peak at 868 cm⁻¹ represents the breathing mode of crown ether-like Li(G4)⁺ 44, which appears at 3.3 V and then shifts to 874 cm⁻¹ as the potential decreases to 2.7 V, as shown in Figure S13” Looking at Figure S13 and Figure 3 – the reviewer does not observe a clear shift in the peak. The data looks too noisy to observe a shift and the peaks could be said to not shift at all.

4. In Figure S14 on graphite similar peaks are observed than that showed in Figure 3, weaker peaks, but some as weak as seen on graphene and similar to 3LG (Figure S15). A weak ca. 930 and 1138 cm⁻¹ could be present at 2.1 V (figure S14). These bands are more distinct in Figure 3. The 1138 cm⁻¹ cannot be observed in Figure 4.

5. Why is the D band visible in 6LG but not graphene? Is it possible to measure 6LG on a pristine basal layer without edge sites?

Minor comment:

“Supercapacitor” is an industry term rather than scientific – change to “electrochemical capacitor” in title and text, particularly as study is fundamental in that it looks at graphene/electrolyte interface rather than device.

Abstract: text unclear – Li(G4)FSI not defined.

The authors claim in abstract “offers fresh insights for optimizing of the energy and power performance of supercapacitors”. This aspect was not clearly shown or articulated in the text. Authors did not show a link from the fundamental study to how performance could be improved and maybe required as section with a paragraph or two drawing together the data and demonstrating how the study impacts on device.

Throughout text: write “behavior” (not “behaviors”) (reads better)

Change coloring in Figure 5 – yellow is barely visible.

Response to Reviewers

We would like to express our appreciation for the reviewers' positive response and valuable comments on our manuscript. And we are grateful to the opportunity given to clarify the concerns raised in the reviewer's report. We hope that our additional experiments and careful replies and revisions adequately address the reviewers' comments and strengthen our revised manuscript. Our responses to the comments are listed point-by-point as follows, and the corresponding changes made in the revised manuscript are highlighted in yellow.

Reviewers' comments and our Response

Reviewer #1 (Remarks to the Author):

In this manuscript the authors designed a gap enhanced Raman spectroscopic strategy to characterize the dynamic interfacial change of graphene to understand the charging mechanism and ion arrangement at the graphene/electrolyte interface. By employing such a strategy combined with complementary characterization techniques, the authors delve into potential-dependent configuration of adsorbed ions and capacitance curves for graphene with various numbers of layers. It was found that the charging mechanism shifts from co-ions desorption on single-layer graphene to co-ions/counter-ions exchange domination on few-layer graphene. Generally, the manuscript is well organized but there are still some concerns which need to be solved, including the logic thread, experimental design and data processing. Thus, I recommend the acceptance after the following comments are considered.

Response: We sincerely thank the reviewer for the constructive comments.

Comment 1: In the abstract or the introduction, the authors did not mention how the transformation of the double layer charging mechanism affects the electrochemical performance. For example, values for the specific capacitance per area under different situations and the possible reasons could be summarized.

Response: We thank the reviewer for this comment. To our knowledge, extensive experimental and theoretical work have revealed three potential mechanisms for double

layer charging storage: counter-ions adsorption, ion exchange, and co-ions desorption. The process of pure counter-ions adsorption could facilitate rapid charging, resembling a front-like movement where ions migrate into the interior of carbon nanopores. In contrast, ion exchange involves ions migrating in opposite directions. These distinct mechanisms lead to variation in ionic density, consequently influencing ion packing during charging and thereby impacting the charging rate (J. Phys. Chem. Lett. 2016, 7, 36-42). Kondrat and Kornyshev highlighted that counter-ions adsorption is impeded by an entropic penalty. Additionally, unfavorable electrostatic (enthalpic) terms arise from the close packing of ions with the same charge. Charging via co-ions desorption is anticipated to alleviate the enthalpic penalty associated with interactions among charges, thereby simultaneously enhancing entropy. Consequently, co-ions desorption should optimize capacitance (Nanoscale Horiz., 2016, 1, 45-52). The ion-exchange mechanism, in principle, mitigates the enthalpic penalty linked to denser ion packing by maintaining a relatively constant total density throughout charging. Simultaneously, the entropic penalty associated with charging is also reduced (J. Am. Chem. Soc. 2016, 138, 5731-5744). We have incorporated additional discussions in the revised manuscript regarding how the transformation of the double layer charging mechanism affects the electrochemical performance (Page 14, Line 371), and we have supplemented the values for the specific capacitance per area under different situations and the reason in the abstract (Page 1, Line 25).

“Charging via co-ions desorption is anticipated to alleviate the enthalpic penalty associated with interactions among like charges, thereby simultaneously enhancing entropy. Consequently, co-ions desorption should optimize capacitance⁵¹. The ion-exchange mechanism, in principle, mitigates the enthalpic penalty linked to denser ion packing by maintaining a relatively constant total density throughout charging. Simultaneously, the entropic penalty associated with charging is also reduced⁵².” (Page 14, Line 371)

“The increase in area specific capacitance from 64 to 145 $\mu\text{F cm}^{-2}$ is attributed to the influence on ion packing, thereby impacting the electrochemical performance.” (Page 1, Line 25)

Comment 2: The full names of the abbreviations appeared for the first time in the abstract such as [Li(G4)] and [FSI] should be given. The reason to choose this electrolyte for the research may be explained.

Response: We thank the reviewer for this comment. We have provided the full name of the abbreviations in the abstract for the first time they appear.

Meanwhile, we have further elaborated on the reason behind the selection of the electrolyte used in this work. We have included the corresponding discussions on Page 3, Line 67 and Page 16, Line 407.

“Lithium bis(fluorosulfonyl) imide (LiFSI) with a weakly associated anion can promote ion dissociation. Glyme molecules ($\text{CH}_3\text{O}(\text{CH}_2\text{-CH}_2\text{O})_n\text{CH}_3$), serving as multidentate ligands, create stable complexes with alkali metal cations in electrolyte. These complexes contribute to high conductivity, low viscosity, and low melting points in the system, thereby further reducing impedance and enhancing rate capability¹⁵.” (Page 3, Line 67)

“LiFSI exhibits weak ion-ion interactions, while G4 provides strong ion-solvent interactions, which collectively influence the solvation process, thus affecting the Li^+ deintercalation behavior on the electrode surface.” (Page 16, Line 407)

Comment 3: In Figure. 1d, how AFM image confirms that the transferred sample is single-layer graphene is not clear enough.

Response: We thank the reviewer for this comment. We have refined the figure to more clearly depict the morphology of the graphene. As shown in Figure R1–1, the region to the left of the red dashed line represents the sapphire substrate, while the region to the right shows the transferred graphene on the sapphire. Sapphire substrate is rather flat, allowing for accurate confirmation of the graphene layer's thickness through the height profile. The measurements confirm a thickness of 0.4 nm, aligning with the expected thickness of single-layer graphene. We have replaced the original Figure 1d with the following updated version.

Figure R1–1, AFM image of single layer graphene on sapphire substrate and sectional analysis of the white line in the image for measuring the height of the transferred graphene. Scale bar: 300 nm.

Comment 4: In Figure. 2b, the numerical value of the current density displayed on the vertical axis is not clear enough. It is suggested to divide it into three charts.

Response: We thank the reviewer for this comment. We have split Figure 2b in the revised manuscript into three separate charts to clearly present the numerical value of current density (as shown in Figure R1–2).

Figure R1–2, Cyclic voltammograms of SLG, 3LG and 6LG electrodes plotted with respect to electrode potentials versus Li^+/Li . Scan rate: 10 mV s^{-1} .

Comment 5: In Figure. 2d and 2e, the authors stated that the thickness in the Columbic ordering layers is 0.52nm and 0.32nm, respectively, but it is difficult to distinguish two peaks from the figures. The positions may be indicated more clearly. The statistical data on multiple graphene samples for the measurement may help to determine the error of values measured.

Response: We thank the reviewer for this comment. We have updated Figures 2d and 2e in the revised manuscript to include 20 force curves at each potential (as shown in Figure R1–3), enhancing the reproducibility of the data. Additionally, the corresponding statistical data and three representative force curves at each potential have been presented in Figure S12–13 of the revised Supplementary Information to clearly differentiate the thicknesses of the two layers. (As shown in Figure R1–4 and R1–5)

Figure R1–3, Two-dimensional graphs of AFM force-distance profiles from 20 force curves under E<PZC and E>PZC.

Figure R1-4, (a) Three AFM force curves when E<PZC. (b) The distribution of the first layer thickness and force value when E<PZC. (c) and (d) Two-dimensional histograms of the first layer thickness and the second layer thickness when E<PZC.

Figure R1-5, (a) Three AFM force curves when E>PZC. (b) The distribution of the first layer thickness and force value when E>PZC. (c) and (d) Two-dimensional histograms of the first layer thickness and the second layer thickness when E>PZC.

Comment 6: In Figure. 3a, the reason for the absence of G and 2D peaks of single-layer graphene shall be explained. Readers may want to know the attribution of the strong vibration peak appearing at 2.9 V at $\sim 1400\text{ cm}^{-1}$.

Response: We thank the reviewer for this comment. The test depicted in Figure 1c of the original manuscript aimed to characterize the successful transfer of graphene onto an Au substrate by monitoring changes in the G and 2D bands. For this purpose, a 532 nm laser wavelength was employed, which offers excellent sensitivity and enables efficient spectral acquisition in a short timeframe, making it well-suited for carbon material analysis. Consequently, the G and 2D peaks of single-layer graphene were clearly observable. However, in Figure 3a of the original manuscript, the experimental conditions were modified to a 785 nm laser wavelength to investigate the adsorption configuration of ions at the graphene/electrolyte electrochemical interface. This wavelength minimizes fluorescence interference from electrolyte (as shown in Figure S21 in the original Supporting Information), resulting in a higher signal-to-noise ratio and improved spatial resolution, albeit with reduced Raman scattering intensity. Under these conditions, the G and 2D peaks were not observed. This is attributable to the lower excitation energy decreases associated with longer laser wavelengths. In the very low energy regime, the intensity of the G and 2D peaks is suppressed due to the conservation of angular momentum associated with continuous rotational symmetry in the low-energy regime. As the excitation energy increases, quantum interference effects become destructive, maintaining low intensity levels and resulting in the cancellation of resonant and non-resonant contributions to the Raman scattering matrix elements. Ultimately, at larger laser energies, resonant contributions from the K-M direction dominate the Raman scattering matrix elements (PHYSICAL REVIEW B 2017, 95, 195422). We have incorporated the corresponding discussions into the revised manuscript (Page 10, Line 267).

Regarding the strong vibration peak appearing at 2.9 V at $\sim 1400\text{ cm}^{-1}$, it is ascribed to the asymmetry stretching of SO_2 . When the potential shifts from negative to positive relative to PZC, the interface undergoes a transition from FSI^- co-ions desorption to Li(G4)^+ co-ions desorption. The alteration in the configuration of FSI^- leads to the

observed SO₂ asymmetry stretching. We have included the corresponding discussion into the revised manuscript (Page 11, Line 302).

“We opted for a 785 nm laser wavelength to investigate the adsorption configuration of ions and avoid fluorescence interference. In the very low energy regime, the intensity of the G and 2D peaks is suppressed due to the conservation of angular momentum associated with continuous rotational symmetry in the low-energy regime⁴³. Consequently, the G and 2D peaks were not observed.” (Page 10, Line 267)

“Regarding the strong vibration peak appearing at 2.9 V at ~1400 cm⁻¹, it is ascribed to the asymmetry stretching of SO₂. When the potential shifts from negative to positive relative to PZC, the interface undergoes a transition from FSI⁻ co-ions desorption to Li(G4)⁺ co-ions desorption. The alteration in the configuration of FSI⁻ leads to the observed SO₂ asymmetry stretching.” (Page 11, Line 302)

Comment 7: In Figure. 3b, different potential regions are not reflected in the figure. It would be better if the authors could provide the values of the potential regions.

Response: We thank the reviewer for this comment. The potential regions have been clearly indicated in Figures 3b and 4b of the revised manuscript, as shown in Figure R1–6 and R1–7.

Figure R1–6, Proposed adsorption configuration of [Li(G4)][FSI] at single-layer graphene in different potential region (including E < PZC, E = PZC and E > PZC).

Figure R1–7, Proposed adsorption configuration of [Li(G4)][FSI] at six-layer graphene in different potential region (including E<PZC, E=PZC and E>PZC).

Comment 8: In Figure. 3a and Figure. 4a, the corresponding meanings of the blue and purple parts in the Raman spectra should be explained in the caption.

Response: We thank the reviewer for this comment. In the revised manuscript, the corresponding meanings of the blue and purple parts in the Raman spectra have been explained in the captions of Figure 3a and Figure 4a.

Comment 9: Is there a connection between the potential-dependent coordination structure of [Li(G4)][FSI] at graphene/electrolyte interface mentioned in the last part and the charge storage mechanism mentioned earlier? Please explain.

Response: We thank the reviewer for highlighting this point. Based on experiments conducted on graphene with varying layers, the potential-dependent coordination structure of [Li(G4)][FSI] at the graphene/electrolyte interface exhibits largely similar behavior, with the exception of the peak of AGG. Therefore, we speculate that the predominant mechanisms in electric-double-layer capacitors are primarily determined by the layers of the graphene, rather than the coordination structure. However, in lithium-ion capacitors, the operational principles of graphene cathodes mirror those of electric-double-layer capacitors. During charge and discharge cycles, ions undergo adsorption and desorption processes via the electrical double layer mechanism. While

at the anode, Li^+ ions undergo intercalation/deintercalation from the electrode material. This process involves the desolvation of cations, which is highly correlated with the interfacial coordination structure. We hope the related findings will be published in a separate paper.

Reviewer #2 (Remarks to the Author):

The authors show that placing graphene and few-layer graphene over a gold substrate and then coupling with SHINERS on the carbon surface– the graphene/ electrolyte interface can be investigated in systems relevant to electrochemical capacitors under potential control. Data shown in Figure 2 is impressive and the authors demonstrate the changing capacitance shape from U to V shape going from SLG, 3LG and 6LG. Force curves show the flipping of the ions either side of pzc. The SHINERS study does show systematic changes observed in a few “weak” peaks. Overall an interesting study – but before acceptance the authors need to clarify some of their interpretation and presentation of the data as some aspects seem unclear.

Response: We sincerely thank the reviewer for the constructive comments.

Comment 1: Is the Au substrate itself SERS active? – authors need to demonstrate lack of or presence of enhancement and provide some characterisation of the Au films.

Response: We thank the reviewer for this comment. For the Au substrate we used, the inherent SERS activity is non-existent. We chose the typical SERS probe molecule, pyridine (Py), to probe the SERS activity of the Au substrate itself. Raman spectra of Py were recorded under the same experimental conditions, yet no discernible characteristic peak corresponding to Py adsorption on Au was observed. Consequently, the Au substrate itself exhibits no enhancement effect. The signals at graphene/electrolyte interface were obtained through a gap-enhanced Raman spectroscopy strategy, employing a sandwich configuration. This approach utilizes the localized surface plasmon (LSP) effect coupling between shell-isolated nanoparticles (SHINs) and metal substrate to produce an enhancement effect. We have included Raman spectra in Figure S5 of the revised Supplementary Information, confirming the absence of Py signals adsorbed on the Au films, as shown in Figure R2–1.

Figure R2–1, Raman spectra of Au film in 0.01 M Py. Laser power was controlled at about 0.7 mW and accumulation time was set to 60 s.

Comment 2: Authors discuss that even with SHINERS no appreciable enhancement can be made on graphene/carbon substrates and only when Au underlayer is present this is possible. Is graphene unique in this sense requiring the Au to couple with? In the authors seminal 2010 Nature paper – they show an example of detection of pesticide on an orange, what is it about an orange substrate whereby SHINERS works, whereby graphene not so?

Response: We thank the reviewer for this comment. We have employed COMSOL finite element method to calculate the enhancement factor between HOPG/graphene and shell-isolated nanoparticles, as shown in Figure R2–2 and R2–3 and Figures S3–4 in the revised Supplementary Information. Our work reveals that the coupling interaction between HOPG/graphene and shell-isolated nanoparticles is weak. The hotspots are primarily localized at the junction between the nanoparticles, away from the probed surface. This hinders the acquisition of high-intensity surface-enhanced Raman signals, highlighting the significance of further coupling with the Au substrate. Additionally, the intensity of surface-enhanced Raman signals is influenced by various factors, including molecular scattering cross-section, refractive index of the substrate surface, and the incident angle of the nanoparticles with respect to the probed surface. These factors collectively impact the interaction between orange and SHINERS, enabling the detection of pesticide in the 2010 Nature paper.

Figure R2–2, Electromagnetic field distribution around Au@SiO₂/HOPG simulated by COMSOL finite element method.

Figure R2–3, Electromagnetic field distribution around Au@SiO₂/graphene/SiO₂ simulated by COMSOL finite element method.

Comment 3: The authors state: “The peak at 868 cm⁻¹ represents the breathing mode of crown ether-like Li(G4)⁺, which appears at 3.3 V and then shifts to 874 cm⁻¹ as the potential decreases to 2.7 V, as shown in Figure S13” Looking at Figure S13 and Figure 3 – the reviewer does not observe a clear shift in the peak. The data looks too noisy to observe a shift and the peaks could be said to not shift at all.

Response: We thank the reviewer for this comment. We reprocessed the spectral peaks of the crown ether-like Li(G4)⁺ breathing mode in the single-layer graphene configuration (Figure S13) to improve signal-to-noise ratio and confirmed a clear shift in the peak. Additionally, we investigated the variation of peak with electrode potential, known as the Stark slope, in the single-layer graphene configuration (as shown in Figure R2–4). We extracted the central peak of the Li(G4)⁺ breathing mode and plotted it as a function of potential (Li/Li⁺). The peak-versus-potential plot can be fitted with a linear equation, with the fitted line representing the slope of the Stark shift in cm⁻¹/V. In the single-layer graphene configuration, the Stark slope was -6 cm⁻¹/V, indicating a

clear peak shift with potential. The change in slope suggests that gap coupling enhances the observation of $\text{Li}(\text{G4})^+$ orientation change on the single-layer graphene surface, thereby providing insights into the charging mechanism at the interface.

Figure R2-4, (a) The spectra of the crown ether-like breathing mode of $\text{Li}(\text{G4})^+$, which were smoothed using second-order processing with LabSpec software. **(b)** The Raman shifts of crown ether-like breathing mode of $\text{Li}(\text{G4})^+$ on single-layer graphene as a function of electrode potential.

Comment 4: In Figure S14 on graphite similar peaks are observed than that showed in Figure 3, weaker peaks, but some as weak as seen on graphene and similar to 3LG (Figure S15). A weak ca. 930 and 1138 cm^{-1} could be present at 2.1 V (figure S14). These bands are more distinct in Figure 3. The 1138 cm^{-1} cannot be observed in Figure 4.

Response: We thank the reviewer for this comment. To better compare the adsorption behavior of different electrolyte components at interfaces, we examined single-layer graphene (Figure 3), three-layer graphene (Figure S15), graphite (Figure S14), and bulk electrolyte at 2.1 V (vs. Li^+/Li). We observed distinct Raman peaks at 930 and 1138 cm^{-1} in the single-layer configuration, indicating $\text{Li}(\text{G4})^+$ adsorption with a flat orientation on the surface, consistent with our analysis in the manuscript. In the three-layer graphene configuration, the intensities of the 930 and 1138 cm^{-1} Raman peaks weakened, indicating a transitional state in the charging mechanism. Conversely, in the graphite configuration, minimal changes in the above Raman peaks were observed,

suggesting the absence of gap coupling enhancement. We have included Raman spectra of the electrolyte adsorbates at the interface in Figure S16 of the revised Supplementary Information (as shown in Figure R2–5). In the six-layer configuration (Figure 4), the charging mechanism converted to ion exchange dominance, $\text{Li}(\text{G4})^+$ demonstrated a faster response to electric field. The CH_2 , CO , and CC groups did not show potential dependence, leading to the absence of the 1138 cm^{-1} Raman peak. Overall, combining experimental data with COMSOL simulation results, the variation in peak intensity among different configurations correlates with the charging mechanism.

Figure R2–5, Raman spectra of the electrolyte adsorbates at the interface for single-layer graphene, three-layer graphene, and graphite configurations at 2.1 V (vs. Li^+/Li).

Comment 5: Why is the D band visible in 6LG but not graphene? Is it possible to measure 6LG on a pristine basal layer without edge sites?

Response: We thank the reviewer for this comment. The D band typically appears around 1350 cm^{-1} and is attributed to the radial breathing mode of symmetric stretching vibrations of sp^2 carbon atoms within the aromatic ring, usually requiring the presence of defects for activation (Chem. Soc. Rev., 2018, 47, 1822-1873). Metal substrates such as copper can dynamically repair imperfect structures, including defects, during the high-temperature growth process of graphene (Nano Lett. 2012, 12, 3936-3940). Consequently, single-layer graphene films grown on metal substrates at appropriate reaction temperatures generally exhibit good crystallinity, with no prominent defect

band observed in Raman spectra. As the number of layers increases, the intensity of the D band rises, indicating the heightened presence of defects within the graphene structure (Science, 2009, 324, 1312-1314; Carbon, 2016, 96, 203-211). Thus, the D band is visible in 6LG. We have measured various positions on 6LG, where the presence of the D band is observable on a pristine basal layer without edge sites.

Minor comment:

Comment 6: “Supercapacitor” is an industry term rather than scientific – change to “electrochemical capacitor” in title and text, particularly as study is fundamental in that it looks at graphene/electrolyte interface rather than device.

Response: We thank the reviewer for the valuable suggestion. We have changed “supercapacitor” to “electrochemical capacitor” in the revised manuscript.

Comment 7: Abstract: text unclear – Li(G4)FSI not defined.

Response: We thank the reviewer for the valuable suggestion. We have provided the full name of Li(G4)[FSI] (lithium bis(fluorosulfonyl) imide in tetraglyme) in the revised abstract.

Comment 8: The authors claim in abstract “offers fresh insights for optimizing of the energy and power performance of supercapacitors”. This aspect was not clearly shown or articulated in the text. Authors did not show a link from the fundamental study to how performance could be improved and maybe required as section with a paragraph or two drawing together the data and demonstrating how the study impacts on device.

Response: We thank the reviewer's valuable suggestion. We have included additional discussions in the revised manuscript regarding how the transformation of the double layer charging mechanism affects the electrochemical performance. Additionally, we have incorporated a section integrating data to demonstrate the comparison of energy density under different dominant mechanisms for electrochemical capacitor. We made the following revision on Page 14, Line 371.

“Charging via co-ions desorption is anticipated to alleviate the enthalpic penalty associated with interactions among like charges, thereby simultaneously enhancing entropy. Consequently, co-ions desorption should optimize capacitance⁵¹. The ion-exchange mechanism, in principle, mitigates the enthalpic penalty linked to denser ion packing by maintaining a relatively constant total density throughout charging. Simultaneously, the entropic penalty associated with charging is also reduced⁵². We conducted a further comparison of the energy density of electrochemical capacitors operating under two predominant mechanisms, as shown in Table S1. Ion exchange plays a significant role in maximizing ion packing, consequently influencing the charging rate and capacitance.” (Page 14, Line 371)

New Supplementary Table 1 | Comparison of the performance of electrochemical capacitor based on graphene electrodes.

Electrode (graphene)	C_m (F g ⁻¹)	E (Wh kg ⁻¹)
single-layer/single-layer	97	34
six-layer/six-layer	220	78

Comment 9: Throughout text: write “behavior” (not “behaviors”) (reads better)

Response: We thank the reviewer for the valuable suggestion. The manuscript has been revised accordingly.

Comment 10: Change coloring in Figure 5 – yellow is barely visible.

Response: We thank the reviewer for the valuable suggestion. Accordingly, we have changed yellow to lavender in Figure 5 for better visualization.

REVIEWERS' COMMENTS

Reviewer #1 (Remarks to the Author):

The current version is OK and can be accepted for publication.

Reviewer #2 (Remarks to the Author):

The authors have addressed my comments and the revised paper can now be considered to be accepted without further modification.